# Annealed ZnO/Al_2_O_3_ Core-Shell Nanowire as a Platform to Capture RNA in Blood Plasma

**DOI:** 10.3390/nano11071768

**Published:** 2021-07-07

**Authors:** Hiromi Takahashi, Takao Yasui, Annop Klamchuen, Narathon Khemasiri, Tuksadon Wuthikhun, Piyawan Paisrisarn, Keiko Shinjo, Yotaro Kitano, Kosuke Aoki, Atsushi Natsume, Sakon Rahong, Yoshinobu Baba

**Affiliations:** 1Department of Biomolecular Engineering, Graduate School of Engineering, Nagoya University, Nagoya 464-8601, Japan; takahashi.hiromi@a.mbox.nagoya-u.ac.jp (H.T.); paisrisarn.piyawan@gmail.com (P.P.); 2Japan Science and Technology Agency (JST), Saitama 332-0012, Japan; 3Institute of Nano-Life-Systems, Institutes of Innovation for Future Society, Nagoya University, Nagoya 464-8601, Japan; anatsume@med.nagoya-u.ac.jp; 4National Nanotechnology Center (NANOTEC), National Science and Technology Development Agency (NSTDA), Pathum Thani 12120, Thailand; annop@nanotec.or.th (A.K.); n.khemasiri@gmail.com (N.K.); tuksadon@nanotec.or.th (T.W.); 5Division of Cancer Biology, Graduate School of Medicine, Nagoya University, Nagoya 466-8560, Japan; kshinjo@med.nagoya-u.ac.jp; 6Division of Neurosurgery, Graduate School of Medicine, Nagoya University, Nagoya 466-8560, Japan; kitano.youtarou@f.mbox.nagoya-u.ac.jp (Y.K.); kaoki@med.nagoya-u.ac.jp (K.A.); 7King Mongkut’s Institute of Technology Ladkrabang, College of Materials Innovation and Technology, Bangkok 10520, Thailand; sakon.ra@kmitl.ac.th; 8Institute of Quantum Life Science, National Institutes for Quantum and Radiological Science and Technology, Chiba 263-8555, Japan

**Keywords:** nanowire structure, RNA analysis, hydrothermal synthesis, nanodevice, nucleic acid analysis

## Abstract

RNA analytical platforms gained extensive attention recently for RNA-based molecular analysis. However, the major challenge for analyzing RNAs is their low concentration in blood plasma samples, hindering the use of RNAs for diagnostics. Platforms that can enrich RNAs are essential to enhance molecular detection. Here, we developed the annealed ZnO/Al_2_O_3_ core-shell nanowire device as a platform to capture RNAs. We showed that the annealed ZnO/Al_2_O_3_ core-shell nanowire could capture RNAs with high efficiency compared to that of other circulating nucleic acids, including genomic DNA (gDNA) and cell-free DNA (cfDNA). Moreover, the nanowire was considered to be biocompatible with blood plasma samples due to the crystalline structure of the Al_2_O_3_ shell which serves as a protective layer to prevent nanowire degradation. Our developed device has the potential to be a platform for RNA-based extraction and detection.

## 1. Introduction

RNAs, including messenger RNAs (mRNAs) and microRNAs (miRNAs), have remarkable roles in numerous biological processes that promoted their service as biomarkers for clinical applications [1,2,3]. Recent technological advancements led to the development of RNA detection technologies for the early detection of cancer [4,5] and of viral infection [6,7]. Currently, there are two types of RNA detection technologies: direct detection [8,9] and nucleic acid amplification-based detection [10,11]. For direct detection, the commonly used methods are northern blotting [12] and fluorescence in situ hybridization [13]; these methods have several drawbacks: they are time-consuming, labor-intensive, and poor in sensitivity. Nucleic acid amplification-based detection is based on reverse transcription PCR (RT-PCR) and other isothermal amplification methods, including loop-mediated isothermal amplification (LAMP) [14] and rolling circle extension-actuated LAMP [15]. However, these techniques require a complex primer design and the conversion of RNA to DNA and subsequent template replication steps. These steps can lead to the loss of sample and the risk of contamination, resulting in false-positive and negative results. In addition, the use of RNAs for clinical applications is still hindered by the diluted concentration of RNAs in body fluids [16]. Therefore, it is essential to develop a platform that can enrich and subsequently detect RNA to prevent those issues.

In recent years, nanostructure-based microfluidic platforms were utilized widely for nucleic acid detection [17,18,19] as nanostructures can enhance detection sensitivity due to their high surface-to-volume ratio. One of the commonly used nanostructures is nanowires because of their: (1) ease of fabrication, ensuring uniformity, reproducibility, and scalability, and (2) ease of modification to make them selective for target biomolecules. Field-effect transistors (FETs)-based nanowire detection is one of the methods for detecting target biomolecules [20,21] through surface modifications using specific bio-receptors. However, this technique has limitations due to its stability under complex biological samples [22]. Thus, a platform that is biocompatible with real biological samples and can concentrate RNA molecules would be beneficial for the application of RNA-based detection. Our previous work showed that the ZnO/Al_2_O_3_ core-shell nanowire enabled the capture of single-stranded DNA with high efficiency [23]. Based on this and the fact that most RNAs are single-stranded nucleic acid, we were motivated to develop the ZnO/Al_2_O_3_ core-shell nanowire device as a platform to capture RNA molecules. In this work, we fabricated ZnO nanowire as the core using the hydrothermal synthesis method, followed by the deposition of Al_2_O_3_ as the shell with the postannealing treatment at 700 °C. The annealed ZnO/Al_2_O_3_ core-shell nanowire showed good biocompatible performance towards blood plasma, and we anticipated that such performance might further facilitate the detection of RNA directly from blood plasma samples, as shown in Figure 1. The developed nanowire was utilized as a platform that could capture RNA molecules and was biocompatible with blood plasma samples. This nanowire also showed versatile application potentials, especially in the context of RNA analytical platforms.

## 2. Materials and Methods

### 2.1. Preparation of Blood Plasma Samples

Whole blood was collected in Streck Cell-Free DNA BCT tube (Guardant Health, Inc., Redwood, CA, USA). To isolate plasma, whole blood was centrifuged (1600× *g* for 10 min at 10 °C), and the resulting supernatant was removed by additional centrifugation (3220× *g* for 10 min at 10 °C). Debris was removed and plasma was transferred every 1 mL to another microtube. Blood plasma sample was stored at −80 °C until use.

### 2.2. Preparation of Nucleic Acid Samples

For cell-free DNA (cfDNA) and genomic DNA (gDNA), A549, lung cancer cell line, was purchased from the American Type Culture Collection (Rockville, MD, USA) and maintained in RPMI-1640 medium (Wako Pure Chemical Industries, Osaka, Japan) supplemented with 5% fetal bovine serum (Thermo Fisher Scientific Inc., Waltham, MA, USA) and antibiotic-antimycotic reagent (Wako Pure Chemical Industries, Osaka, Japan) at 37 °C in a humidified incubator with 5% CO_2_. Genomic DNA (gDNA) from the cell lines was extracted using a standard phenol-chloroform method. gDNA were fragmented to an average size of 200 and 2000 bp, using Covaris S220 manufactured by Covaris (Woburn, MA, USA) and used as cfDNA and gDNA, respectively. mRNA (150 bp) from the cell lines was extracted using TRIzol reagent (Thermo Fisher Scientific Inc., Waltham, MA, USA) following the manufacturer’s protocol. For miRNA, 3 different kinds of miRNA (-21, -155, -124) were purchased from the manufacturer (Thermo Fisher Scientific Inc., Waltham, MA, USA) (miR-21, UAGCUUAUCAGACUGAUGUUGA; miR-155, UUAAUGCUAAUCGUGAUAGGGGUU; miR-124, CGUGUUCACAGCGGACCUUGAU).

### 2.3. Fabrication of ZnO/Al_2_O_3_ Core-Shell Nanowire

The fabrication method of the nonannealed ZnO/Al_2_O_3_ core-shell nanowire was described in the previous study [23]. For the annealed ZnO/Al_2_O_3_ core-shell nanowire, the nonannealed nanowire was annealed at 700 °C using KDF-75 (Denken–Highdental, Kyoto, Japan). Next, a PDMS microchannel (width, 20 mm; length, 20 mm; and height, 25 μm) with a 10-μm-high herringbone structure was sealed on the ZnO/Al_2_O_3_ core-shell nanowire substrate using a soft plasma etching system (SEDE-PFA, Meiwafosis, Tokyo, Japan). Finally, the PDMS microchannel was punctured with a 0.5 mm UNICORE (Harris, GE Healthcare, Bucks, UK) to form holes as an inlet and an outlet. A PEEK tube (ICT-55P, Institute of Microchemical Technology Co., Ltd., Kanagawa, Japan) was inserted into each hole.

### 2.4. Structural Characterization

Morphological characterization of the nanowire was carried out using SEM (Supra 40VP, Carl Zeiss, Jena, Germany), operated at an acceleration voltage of 5 kV. The size range of the nanowire was measured by ImageJ software. To investigate the cross-sectional structure of the ZnO/Al_2_O_3_ core-shell nanowire, an ultramicrotome cutting process was utilized to prepare an ultrathin section (<100 nm) ZnO/Al_2_O_3_ core-shell nanowire. First, the ZnO/Al_2_O_3_ core-shell nanowire was cut from the substrate and then embedded in epoxy resin (Epon 812, Electron Microscopy Science, Hatfield, PA, USA). Ultrathin sections were cut using an ultramicrotome (EM UC7, Leica, Vienna, Austria) with a 45° diamond knife (DiATOME, Hatfield, PA, USA). Samples were sectioned and collected on copper grids. Bright-field TEM and EDX were performed on a JEOL 2100 (JEOL USA Inc., Peabody, MA, USA), operated at 200 kV. Crystal structure of the prepared ZnO/Al_2_O_3_ core-shell nanowire with and without postannealing treatment was examined by means of X-ray diffraction (XRD) (Rigaku Smartlab, Tokyo, Japan) using Cuk_α_ radiation (λ = 0.154 nm).

### 2.5. Capture Experiment on a Nanowire for Circulating Nucleic Acids

50 µL of 1 ng/µL of each nucleic acid, referred to as the introduced sample hereafter, was supplied into a ZnO/Al_2_O_3_ core-shell microfluidic device using a syringe pump with the flow rate of 5 µL/min. The sample passed through the outlet was collected and is referred to as the collected sample. Next, 50 µL of distilled water was introduced into the device to wash out noncaptured DNAs and is referred to as the noncaptured sample. The concentrations of the introduced sample, collected sample, and noncaptured sample were quantified by qPCR.

### 2.6. Real-Time Quantification of Nucleic Acid

For cfDNA and gDNA quantification, the qPCR reaction mixture contained 2 µL of cfDNAs, 5 µL of TaqMan Gene Expression Master Mix (Thermo Fisher Scientific Inc., Waltham, MA, USA), 2.75 µL of distilled water, and 0.25 µL of primer GAPDH housekeeping genes (forward, 5′-CCTCCCGCTTCGCTCTCT-3′; and reverse, 5′-GGCGACGCAAAAGAAGATG-3′). All reactions were performed with an initial denaturation step for 2 min at 50 °C, then 10 min at 95 °C followed by 50 cycles of 95 °C for 15 s, and ending with an annealing at a temperature of 60 °C for 1 min. For mRNA quantification, 3.5 µL of RNA sample was first treated in the reverse transcription (RT) step for cDNA synthesis using the PrimeScript^TM^ Reagent Kit (Perfect Real Time) (Takara Bio Inc., Shiga, Japan) and 100 µM Random Primer (Random Primer (hexadeoxyribonucleotide mixture; pd (N)6) (Takara Bio Inc., Shiga, Japan). All steps were performed following the manufacturer’s protocol. The RT reactions were carried out at 37 °C for 15 min and 85 °C for 3 s, then held at 4 °C using the thermal cycler (Bio-Rad Laboratories Inc., Hercules, CA, USA). For the qRT-PCR step, the reaction mixture contained 1.75 µL of cDNA, 5 µL of TB Green^TM^ Premix Ex Taq^TM^ II (Tli RNaseH Plus) (Takara Bio Inc., Shiga, Japan), 2.75 µL of distilled water, and 0.25 µL of primer; GAPDH housekeeping genes. All reactions were performed with an initial denaturation step for 30 s at 95 °C, followed by 40 cycles of 95 °C for 3 s, and ending with annealing at a temperature of 60 °C for 30 s. For miRNA quantification, 5 µL of miRNAs, hsa-miR-21-5p, was first treated by the reverse transcription (RT) step for cDNA synthesis using TaqMan MicroRNA Reverse Transcription Kit (Thermo Fisher Scientific Inc., Waltham, MA, USA). All steps were performed following the manufacturer’s protocol. The RT reactions were carried out for 30 min at 16 °C, followed by 30 min at 42 °C, and 5 min at 85 °C, then holding at 4 °C using the thermal cycler. For the qRT-PCR step, the reaction mixture contained 0.67 of µL of cDNA, 0.5 µL of 20X TaqMan^TM^ Small RNA Assay for hsa-miR-21-5p (Thermo Fisher Scientific Inc., Waltham, MA, USA), 5 µL of TaqMan Fast Advanced Master Mix (Thermo Fisher Scientific Inc., Waltham, MA, USA), and 3.84 µL of Nuclease–free water. All reactions were performed with an initial denaturation step for 20 s at 95 °C, followed by 40 cycles of 95 °C for 1 s, and annealing at a temperature of 60 °C for 20 s. All qPCR reactions were performed with the Applied Biosystems QuantStudio 3 Real-Time PCR System (Thermo Fisher Scientific Inc., Waltham, MA, USA). The concentration of an unknown sample was calculated using a calibration curve constructed with known concentration samples.

## 3. Results

### 3.1. Compatibility of the Nanowire with Blood Plasma Samples

We first examined biocompatibility of the nanowire structure with blood plasma samples. To do this, we supplied a blood plasma sample into both nonannealed and annealed ZnO/Al_2_O_3_ core-shell nanowire devices to observe biodegradability and biocompatibility. Before and after supplying blood plasma samples to each device, we compared both annealed and nonannealed ZnO/Al_2_O_3_ core-shell nanowire structure using scanning electron microscopy (SEM). SEM images showed that the nonannealed ZnO/Al_2_O_3_ core-shell nanowires dissolved within 5–10 min after supplying the blood plasma sample, as shown in Figure 2a,b. In contrast, SEM images in Figure 2c,d of the annealed ZnO/Al_2_O_3_ core-shell nanowires showed they were not degraded. Thus, our results confirmed that the annealed ZnO/Al_2_O_3_ core-shell nanowire structure was suitable as a platform for capturing RNA from blood plasma samples.

### 3.2. Morphological and Structural Analyses

Due to their biocompatibility with blood plasma samples, we next made morphological and structural analyses of the annealed ZnO/Al_2_O_3_ core-shell nanowires using various electron microscopy methods. Figure 3a shows the typical cross-sectional (tilt angle = 40°) SEM images confirmed the morphology of the annealed ZnO/Al_2_O_3_ core-shell nanowire, showing that nanowires grew perpendicular to the substrate with a diameter of 90 nm and length of 1380 nm. A low-magnification TEM image in Figure 3b revealed a shell thickness of 9 nm of Al_2_O_3_ was deposited on ZnO nanowire. Next, a high-magnification TEM image in Figure 3c showed the crystalline structure of both ZnO and Al_2_O_3_ with the lattice spacing value of 0.287 nm for ZnO nanowire core and 0.238 nm for Al_2_O_3,_ which corresponded to the orientation growth of (100) ZnO and (110) Al_2_O_3_, respectively. According to these results, the biocompatibility of the annealed ZnO/Al_2_O_3_ core-shell nanowire structure with blood plasma samples was presumably due to the crystalline structure of the Al_2_O_3_ shell as a protective layer to prevent the dissolution. The electron diffraction (ED) pattern had a set of spots for the single-crystal hexagonal ZnO nanowire core and Al_2_O_3_ shell layer, as shown in Figure 3d, and the EDX mapping in Figure 3e–i showed the annealed ZnO/Al_2_O_3_ core-shell nanowire from which we identified the chemical composition of the annealed ZnO/Al_2_O_3_ core-shell nanowire.

Moreover, to confirm the crystalline structure of the annealed ZnO/Al_2_O_3_ core-shell nanowire, we utilized XRD measurement to confirm the nanowire structure. The XRD pattern exhibited out-of-plane X-ray diffractograms (theta-2 theta geometry) of the nonannealed and annealed ZnO/Al_2_O_3_ core-shell nanowire structures (as illustrated in Figure 3i,j). For the nonannealed one, the diffracting patterns at ~34.45° and ~47.58° were attributed to (002) and (102) planes of the hydrothermal ZnO nanowire with the hexagonal wurtzite structure [24]. The predominant intensity of (002) plane strongly suggested that ZnO nanowire hydrothermally grew on the substrate surface with the preferential orientation along [0001] direction (c-axis) [25]. The additional diffraction patterns at ~31.62° and ~37.31° corresponding to (220) and (311) planes of gamma Al_2_O_3_ phase were distinctly observed from the annealed ZnO/Al_2_O_3_ core-shell nanowire [26,27,28]. From the TEM observation and XRD measurement, we concluded that the crystalline structure of the annealed ZnO/Al_2_O_3_ core-shell nanowire provided the biocompatibility with blood plasma samples.

### 3.3. Nucleic Acid Capture on Annealed ZnO/Al_2_O_3_ Core-Shell Nanowire Structure

To investigate the feasibility of the annealed ZnO/Al_2_O_3_ core-shell nanowire structure to capture RNAs, each type of nucleic acid sample was separately supplied into a nanowire microfluidic device using the syringe pump system. Figure 4a shows that both miRNAs and mRNAs showed capture efficiencies on the annealed ZnO/Al_2_O_3_ core-shell nanowire of 98% and 87.5%, respectively. In contrast, cfDNAs (200 bp) and gDNAs (2000 bp) were captured on the annealed ZnO/Al_2_O_3_ core-shell nanowire at lower efficiencies of 28% and 10%, respectively. It was previously reported that the interaction between single-stranded DNA and nonannealed ZnO/Al_2_O_3_ core-shell nanowire may occur via electrostatic interaction due to the zeta potential value [23]. In this work, however, both cfDNAs and gDNAs with the double-stranded structure had the lowest capture efficiencies by the annealed ZnO/Al_2_O_3_ core-shell nanowire. This indicated that the interaction mechanism between the annealed ZnO/Al_2_O_3_ core-shell nanowire and a single-stranded nucleic acid did not occur via the electrostatic interaction from phosphate groups, but by nucleobases through hydrogen bonding. Moreover, the higher capture efficiency of cfDNAs, which have a shorter length than gDNAs, was possibly due to the conformation of gDNAs which might not suitable for capture on nanowire surfaces. Next, we saw that the capture efficiency of miRNAs was saturated starting from 1 pM (as illustrated in Figure 4b). To further demonstrate the capability of the nanowire to capture other types of miRNAs, we supplied three different types of miRNAs (-21, -155, -124), and our developed nanowire devices could capture all these types of miRNAs with high efficiency (as illustrated in Figure 4c). These results suggested that the annealed ZnO/Al_2_O_3_ core-shell nanowire could serve as a platform to capture RNAs, facilitating the development of RNA-based on-chip analysis. However, the limitation exists in the present work, which is to selectively capture a desired target nucleic acid. Further work should be done to expand its specificity for further applications, such as surface modification using sequence-specific oligonucleotide probe.

## 4. Conclusions

The annealed ZnO/Al_2_O_3_ core-shell nanowire device can serve as an RNA analytical platform for biological samples; however, further studies looking at the interaction mechanisms between ZnO/Al_2_O_3_ core-shell nanowires and nucleic acids are essential for future development. We believe that our developed device will offer new opportunities as an on-chip detection platform for RNA-based analysis.

## Figures and Tables

**Figure 1 nanomaterials-11-01768-f001:**
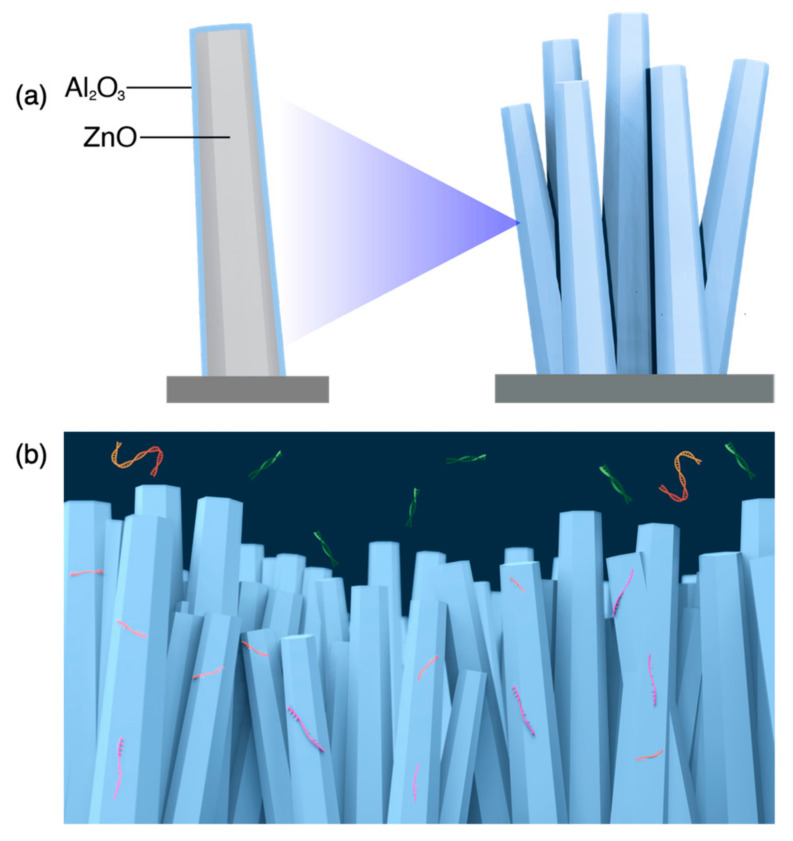
Schematic illustrations. (**a**) ZnO/Al_2_O_3_ core-shell nanowire. (**b**) Capture of miRNAs and mRNAs on nanowire structures; other nucleic acids are not captured.

**Figure 2 nanomaterials-11-01768-f002:**
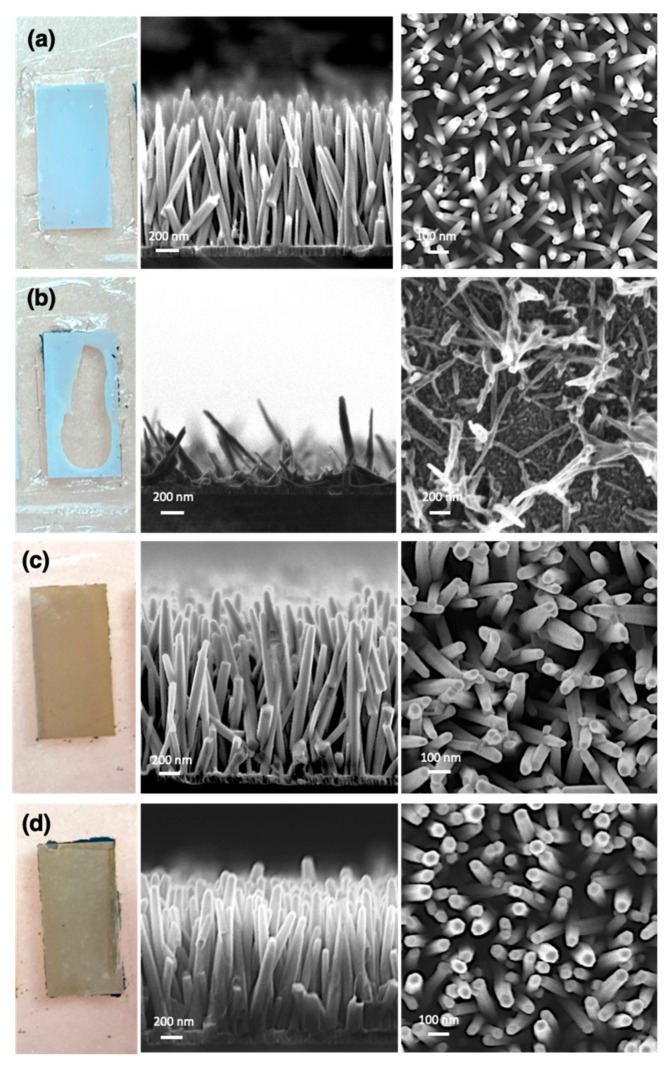
Photos and SEM images. (**a**) Before and (**b**) after supplying blood plasma to the nonannealed ZnO/Al_2_O_3_ core-shell nanowires. (**c**) Before and (**d**) after supplying blood plasma to the annealed ZnO/Al_2_O_3_ core-shell nanowires.

**Figure 3 nanomaterials-11-01768-f003:**
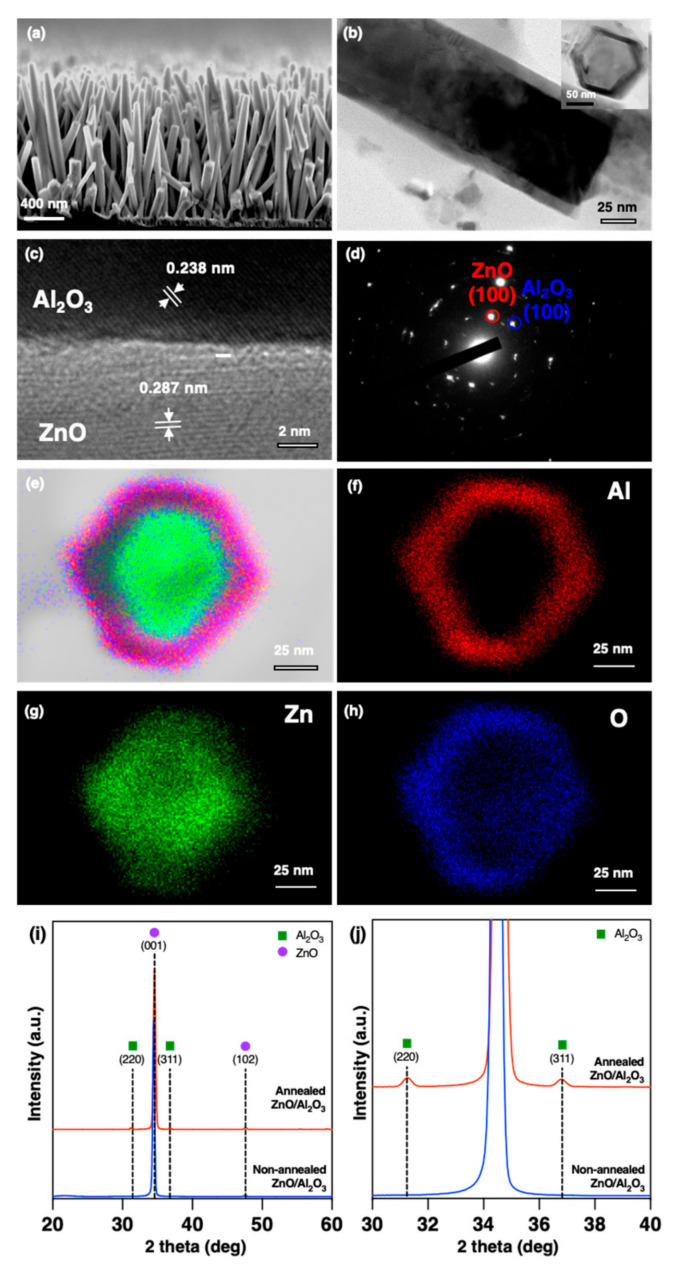
Characterization of ZnO/Al_2_O_3_ core-shell nanowires with postannealing treatment process. (**a**) SEM image and (**b**) TEM of ZnO/Al_2_O_3_ core-shell nanowire. Inset in Figure 3b is cross-sectional TEM image obtained by a nanowire prepared by microtome cutting process (**c**) Atomic-resolution TEM image of ZnO/Al_2_O_3_ core-shell nanowire showing orientation growth of (100) ZnO and (110) Al_2_O_3_, respectively. (**d**) ED pattern of ZnO/Al_2_O_3_ core-shell nanowire. (**e**–**h**) EDX mapping showing core-shell structure of ZnO/Al_2_O_3_ core-shell nanowire. (**i**) XRD patterns of ZnO/Al_2_O_3_ core-shell nanowire with and without postannealing treatment process and (**j**) enlarged XRD patterns of Al_2_O_3_ peak.

**Figure 4 nanomaterials-11-01768-f004:**
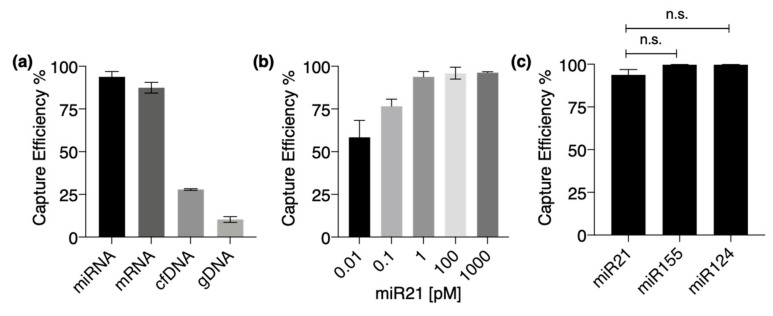
Capture of miRNAs on annealed ZnO/Al_2_O_3_ core-shell nanowires. (**a**) Comparison of capture efficiency of the nucleic acids, miRNAs, mRNAs, cfDNAs, and gDNAs. (**b**) Capture efficiency of miR21 on nanowire at presences of 0.01, 1, 10, and 1000 pM. (**c**) Comparison of capture efficiency of miR21, miR155, and miR124. Error bars show standard deviation for a series of measurements (n = 3). n.s. indicates not significant.

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
