# Peer review of "Annealed ZnO/Al2O3 Core-Shell Nanowire as a Platform to Capture RNA in Blood Plasma"

_nanomaterials, 2021, doi:10.3390/nano11071768_

Round 1
Reviewer 1 Report
The present study demonstrated that when previously developed ZnO/Al2O3 core-shell nanowires were further annealed, these wires can capture RNAs effectively. And wires were resistant against incubation in blood plasma. The authors clearly showed that only annealed wires were stable in blood plasma. This is clear improvement from their previous wire structures (ref. 20), which effectively captured single stranded DNA. Thus it is not surprising that annealed wires also prefer to capture RNAs rather than double stranded DNAs such as cfDNA and gDNA. A major concern is that RNA capture in blood plasma is not demonstrated in this work despite of the claim “a platform to capture RNA in blood plasma”. Therefore, I will recommend publication of this work in Nanomaterials only after addressing following issues.
Comments:
- As mentioned above, the RNA capture ability of the wire in blood plasma must be investigated to validate the claim “a platform to capture RNA in blood plasma”.
- To properly evaluate relative RNA and DNA capture efficiencies, their sizes must be indicated, particularly mRNA, cfDNA, and gDNA.
- In addition, single stranded DNA must also be tested to compare with single stranded RNAs.
Reviewer 2 Report
The manuscript «Annealed ZnO/Al2O3 core-shell nanowire as a platform to capture RNA in blood plasma» by H. Takahashi et al. concerns an important problem of biomedicine – namely, the development of novel nanowire-based diagnostic systems for the detection of microRNAs, cfDNAs, and gDNAs. Attaining good reproducibility, stability of the structures developed in biological fluids, and concentrating nucleic acid molecules on the nanowire chip surface represent very important points in the development of such systems. And this is why the study by Takahashi et al. Is quite actual and interesring. The authors have developed ZnO/Al2O3 nanowire structures, and have demonstrated their stability in a biological fluid. The authors have demonstrated that these structures are able to concentrate nucleic acids on their surface. Takahashi et al. have prepared a high-quality manuscript, which can be accepted just after several minor points, listed below, will be improved.
- The authors have noted that the platform proposed can be employed for the enrichment of RNAs presenting in blood plasma. In this connection, the authors are encouraged to present data on the use of the structures proposed for capturing RNAs from blood plasma mentioned in the Abstract.
- In the Introduction, the authors have considered RNAs detection technologies for the diagnosis of cancer. At the same time, however, the authors should additionally review the technologies employing nanowire detectors, which are based on field-effect transistors – including the papers by Lieber et. al., Ivanov et. al., Stern et. al., Guo X. et al. and other – since the technologies described in these papers allow one to perform direct detection of proteins and RNAs.
Reviewer 3 Report
Comments:
- Please input chemical/reagent seller information in details by including location and country name.
- Section 2.1 last sentence: Was stored (not was store), please check similar typos or grammatical errors.
- How do you confirm the non-toxicity of your materials or characteristics that don’t alter the composition or function of blood plasma? Additional experiment is required to validate it.
- If the plasma sample contains a mixture of miRNA, mRNA, cfDNA or gDNA, how do you capture only specific desired sequence?
- The interaction properties of ZnO/Al2O3 nanowire with RNA or DNA are not determined in the manuscript. However, it is recommended to perform Atomic Force Microscopy (AFM) to visualize the nucleic acid and nanowire interaction surface and alignment which was not possible to observe from SEM.
Round 2
Reviewer 1 Report
The authors addressed several raised issues.
Reviewer 3 Report
The authors did the suggested corrections and the explanations were satisfactory. Thus, the revised paper could be accepted now for the publication in Nanomaterials. Thanks.